# Polypeptide *N*-acetylgalactosaminyltransferase-Associated Phenotypes in Mammals

**DOI:** 10.3390/molecules26185504

**Published:** 2021-09-10

**Authors:** Kentaro Kato, Lars Hansen, Henrik Clausen

**Affiliations:** 1Department of Eco-Epidemiology, Institute of Tropical Medicine, Nagasaki University, 1-12-4 Sakamoto, Nagasaki 852-8523, Japan; 2School of Tropical Medicine and Global Health, Nagasaki University, 1-12-4 Sakamoto, Nagasaki 852-8523, Japan; 3Copenhagen Center for Glycomics, Department of Cellular and Molecular Medicine, Mærsk Building, University of Copenhagen, Blegdamsvej 3B, 2200 Copenhagen N, Denmark; hclau@sund.ku.dk

**Keywords:** polypeptide *N*-acetylgalactosaminyltransferase, UDP-GalNAc: polypeptide *N*-acetylgalactosaminyltransferase, GalNAc-T, *GALNT*, *O*-glycosylation

## Abstract

Mucin-type *O*-glycosylation involves the attachment of glycans to an initial *O*-linked *N*-acetylgalactosamine (GalNAc) on serine and threonine residues on proteins. This process in mammals is initiated and regulated by a large family of 20 UDP-GalNAc: polypeptide *N*-acetylgalactosaminyltransferases (GalNAc-Ts) (EC 2.4.1.41). The enzymes are encoded by a large gene family (*GALNT*s). Two of these genes, *GALNT2* and *GALNT3*, are known as monogenic autosomal recessive inherited disease genes with well characterized phenotypes, whereas a broad spectrum of phenotypes is associated with the remaining 18 genes. Until recently, the overlapping functionality of the 20 members of the enzyme family has hindered characterizing the specific biological roles of individual enzymes. However, recent evidence suggests that these enzymes do not have full functional redundancy and may serve specific purposes that are found in the different phenotypes described. Here, we summarize the current knowledge of *GALNT* and associated phenotypes.

## 1. Introduction

Initiation of mucin-type *O*-glycosylation is controlled by a large family of UDP-GalNAc: polypeptide *N*-acetylgalactosaminyltransferases (EC 2.4.1.41; also referred to as GalNAc-Ts, ppGalNAc-Ts, and ppGaNTases) in animals. These enzymes catalyze formation of GalNAcα1-serine (Ser)/threonine (Thr) linkages in glycoproteins. In humans, 20 isoforms of GalNAc-Ts have been identified and several reviews have described these enzymes in detail [1,2,3]. The importance of each isoform is unclear because phenotypes lacking one or two isoforms have not been observed in experimental animals, except in *Drosophila* [4,5,6,7,8]. However, more recent studies have described *GALNT* mutation-related phenotypes in mammals. The purpose of this review is to summarize these phenotypes and to correct unintentional errors and nomenclature in the literature.

## 2. *GALNT*-Associated Phenotypes in Mammals

### 2.1. GALNT1-Related Phenotypes

The first molecular cloning and characterization were performed for GalNAc-T from bovine tissue [9,10]. Based on the bovine *GALNT1* sequence, rat [11], porcine [12,13] and human [14,15] genes were identified. Murine *Galnt1* is unpublished, but was directly submitted to GenBank by Hagen (GenBank: U73820.1). The amino acid sequence of murine GalNAc-T1 was published in 1997 [16]. *GALNT1* and its protein are ubiquitously expressed in tissues and cell types at high levels [9,10,16,17,18]. This high and broad expression among tissues suggests that disruption of the gene would severely affect organisms. However, disruption of murine *Galnt1* is not fatal and does not cause infertility, although *Galnt1* deficiency does have moderate lethality and one-fourth of homozygous null mice, but not heterozygous mice, die in utero beyond E12.5 or before the age of one month for unknown reasons [19]. The moderate lethality observed in *Galnt1* null mice might indicate that other GalNAc-Ts could partially compensate for GalNAc-T1.

In 2007, utilizing the Cre-loxP recombination system, Tenno et al. showed altered innate and adaptive immune cell trafficking in *Galnt1*-inactivated mice. This correlates with decreased expression of E- and P-selectin ligands on neutrophils and decreased expression of L-selectin ligands on lymph node high endothelial venules (HEVs). Increased apoptosis of B cells in the germinal center, resulting in impaired IgG production, also occurs in GalNAc-T1-deficient mice. A bleeding disorder was also observed in the mice, due to reduction of plasma levels of coagulation factors V, VII, VIII, IX, X and XII, even though platelet homeostasis was unaffected [20]. Immunological impairment of *Galnt1*-deficient mice has also been reported [21]. P-selectin-dependent and E-selectin-mediated leukocyte rolling were significantly impaired and L-selectin-dependent leukocyte rolling was completely abolished in *Galnt1*-deficient mice. P-selectin glycoprotein ligand-1 (PSGL-1) has been suggested to be a target protein for GalNAc-T1 *O*-glycosylation.

Loss of *Galnt1* affects the early stages of murine organogenesis by disrupting secretion of components, especially laminin and collagen IV, of the basement membrane, which is a specialized extracellular matrix (ECM) that is important for mammalian development. Reduced cell proliferation and embryonic submandibular gland (SMG) growth, resulting in decreased integrin and FGF signaling, occur in *Galnt1*-deficient mice. Interestingly, induction of endoplasmic reticulum (ER) stress and the unfolded protein response (UPR), distinct from those observed in *N*-linked glycan deficiency, were observed in SMGs of these mice [22]. The same group also found impaired cardiac function in *Galnt1*-deficient adult mice. In this case, aberrant valve formation caused by increased cell proliferation of developing hearts resulted in aortic and pulmonary valve stenosis and regurgitation, left ventricular hypertrophy, cardiac dilation and valve thickening. In the developing valve tissues of *Galnt1*-deficient mice, transforming growth factor (TGF)-beta associated bone morphogenetic protein (BMP) signaling and mitogen-activated protein kinase (MAPK) signaling were increased. As an antagonist to BMP signaling, epidermal growth factor receptor (EGFR) signaling was decreased in these tissues. These alterations of signaling pathways resulted in increased cell proliferation and larger heart valves in the early stages of development. Expression of ADAMTS (a disintegrin and metalloproteinase with thrombospondin motifs) proteases, ADAMTS1 and ADAMTS5, which have important roles in valvulogenesis, was reduced at the protein level in developing valves of *Galnt1*-deficient mice, resulting in reduction of proteoglycan versican processing. For ECM proteins involved in cardiac development, reduced expression of cartilage link protein 1 (Crtl1) and increased accumulation of collagen I and fibronectin occurred in the mice. Unlike in embryonic SMGs of *Galnt1*-deficient mice, there was no ER stress in the developing heart tissue [19]. These observations are interesting, but the target proteins and underlying mechanisms in the *GALNT1*-related phenotypes are still unclear.

### 2.2. GALNT2-Related Phenotypes

The second GalNAc transferase, GalNAc-T2, was identified and purified from human placenta [15]. Based on the amino acid sequence, polymerase chain reaction (PCR) primers were generated and *GALNT2* was obtained from a cDNA library of the gastric tumor cell line MKN45. *Galnt2* and its protein are ubiquitously expressed in tissues and cell types at high levels in mice [18]. Recent genome-wide association studies (GWASs) have revealed associations of single nucleotide polymorphisms (SNPs) of *GALNT2* with levels of plasma high-density lipoprotein cholesterol (HDL-C) and triglycerides (TG) in humans [23,24,25,26]. Teslovich et al. further showed that liver-specific overexpression of *Galnt2* using an adeno-associated virus (AAV) vector expression system reduced HDL-C levels, while shRNA-mediated knockdown of hepatic *Galnt2* raised HDL-C in mice [26]. These results are supported by a study showing that a GalNAc-T2^D314A^ (p.[Asp314Ala]) heterozygous mutation raised plasma HDL-C levels in humans. This may be due to a loss-of-function (LOF) in GalNAc-T2, caused by the mutation reducing *O*-glycosylation and activity of Apo-CIII, a protein that inhibits lipoprotein lipase (LPL), with a consequent increase in HDL-C [27].

In contrast, other GWAS and in vitro studies have shown that a modest increase in *GALNT2* expression increases HDL-C levels [28]. Furthermore, two cases of human homozygous LOF mutations in hepatic *GALNT2* with low levels of HDL-C and moderate reduction of TG in plasma were recently reported while searching for causes of heritable neurological traits and intellectual disability [29]. These mutations are a T>C in exon 3 of *GALNT2* (p.[Phe104Ser]) and a nonsense mutation in exon 9 (p.[Gln289*]). Both are within a catalytic domain or between the catalytic domains of GalNAc-T2, leading to expression of a non-functional protein [29].

An explanation of the contrasting results among these studies was suggested by Khetarpal et al; overdose of AAV and AAV shRNA used in the earlier study caused GalNAc-T2-independent reduction and shRNA sequence-specific elevation in HDL-C levels, respectively [29]. For the mutated GalNAc-T2^D314A^ enzyme, normal activity of the protein was confirmed [30] and a GalNAc-T2^D314A^ homozygote had low HDL-C levels in humans, indicating that the higher HDL-C in heterozygotes in the earlier study was *GALNT2*-independent [29].

The major target proteins involved in these human phenotypes are angiopoietin-like protein 3 (ANGPTL3), apolipoprotein C-III (ApoC-III) and phospholipid transfer protein (PLTP) [29,31,32]. GalNAc-T2 selectively glycosylates Thr226 of ANGPTL3 adjacent to the furin-like proprotein convertase processing site, and *O*-glycosylation of this site blocks protein cleavage in vitro. Cleavage of ANGPTL3 at this site leads to activation of the protein [33] as an inhibitor of LPL [34] and endothelial lipase (EL) [35]; therefore, LOF of GalNAc-T2 will increase ANGPTL3 activity, lower LPL activity and reduce HDL-C levels. This has been shown clearly in cultured cells and in mice [36]. GalNAc-T2 glycosylates Thr74 of ApoC-III, and if *O*-glycosylation stabilizes the protein, elevated Apo-CIII will lower LPL and increase plasma TG, although species-specific regulation of plasma TG levels has also been reported [29,36,37,38]. The effect of *O*-glycosylation on Thr74 of ApoC-III clearly requires further study, because the absence of *O*-glycosylation at Thr74 did not affect Apo-CIII secretion or binding affinity for lipoprotein in vitro [39], and resulted in normal TG, HDL and ApoC-III levels in humans [40].

Plasma PLTP activity is negatively associated with HDL-C levels and positively associated with TG levels [41]. PLTP activities are partially lowered in rodents and in humans, even though protein expression in plasma was not affected by *GALNT2* deficiency [29]. This partial inhibition of activity can be explained by the ability of another major GalNAc-T expressed in liver—GalNAc-T1—to glycosylate the protein, although not as efficiently as GalNAc-T2 [29,31].

There are many players among lipases and in lipoprotein metabolism, and molecular regulation of lipoproteins is still not fully understood [42]. However, it is clear that *GALNT2* deficiency lowers HDL-C levels and affects lipoprotein metabolism in rodents, nonhuman primates and humans. Novel congenital disorders with global development delay, intellectual disability and decreased HDL-C caused by *GALNT2* deficiency in humans have recently been reported [43], and these multiple phenotypes also occur in *Galnt2*-deficient rodent models. This indicates that LOF of GalNAc-T2 affects the functions of protein substrates in different tissues. A summary of published *GALNT2* mutations and associated phenotypes is shown in Table 1.

### 2.3. GALNT3-Related Phenotypes

Human *GALNT3* was identified using the common sequences of *GALNT1* and *GALNT2* [44], and murine *Galnt3* was identified from mouse testis [45]. In contrast to *GALNT1* and *GALNT2*, expression of *GALNT3* is highly regulated. Human *GALNT3* is mainly found in the pancreas and testis, with lower expression in the kidney, prostate and intestine. Murine *Galnt3* is highly expressed in the testis and moderately in other organs, including the sublingual gland, uterus, cervix, stomach, colon, kidney, submandibular gland and parotid gland [44,45].

Topaz et al. first reported *GALNT3* mutation-associated hyperphosphatemic familial tumoral calcinosis (HFTC) cases in humans [46], and phenotype-related *GALNT3* mutations have subsequently been reported in 62 cases from 39 families. Some of these cases were also summarized by Sun et al. [47] and Rafaelsen et al. [48]. Hyperostosis-hyperphosphatemia syndrome (HHS) is another clinical feature of *GALNT3* deficiency; however, as concluded by Ichikawa et al. [49], we consider HFTC and HHS to be a continuous spectrum of the same disease and we describe both conditions as HFTC in this review. The mechanism underlying HFTC caused by *GALNT3* mutation is a lack of *O*-glycosylation on Thr178 in a furin-like proprotein convertase processing site of phosphaturic factor FGF23; cleavage leads to inactivation of the factor [50].

The 62 reported HFTC cases in 39 families are summarized in Table 2. The disease is caused by both compound heterozygous and homozygous mutations of *GALNT3*. As shown in Table 2, there is no apparent difference in ethnicity, with 11 different ethnicities and a group of unknown ethnicities. A total of 36 different *GALNT3* mutations are represented in the families, of which 16 are missense mutations, 6 are stop mutations, 9 are insertions or deletions causing a frameshift and 5 are splice site mutations (Table 3). The mutations are in all exons, except for exon 1 and exon 11. Exon 1 is a non-coding region of *GALNT3* (NM_004482.4) and exon 11 contains the γ region of the lectin domain (Figure 1). In some previous studies, exon 1 has been skipped in numbering of exons and introns of *GALNT3*, but in this review, exon 1 is included in this numbering. The HFTC-related *GALNT3* mutations are found in all domains, including the initiating ATG start codon, stem region, catalytic domain, linker region and lectin domain of the enzyme. Two of six cysteine residues involved in disulfide bridges in lectin domains of GalNAc-T3 are affected by missense mutations (p.[Cys561Ser] and p.[Cys574Gly]).

Other missense mutations in the catalytic and lectin domains of GalNAc-T3 are potential hypomorphic mutations. A missense mutation in the ATG start codon (p.[Met1Lys]) leads to complete loss of the entire GalNAc-T3 protein, while stop and frameshift mutations cause LOF of the catalytic or lectin domains. The lectin domain of GalNAc-T3 is required for *O*-glycosylation on Thr178 of FGF23 [51], in agreement with homozygous mutations in this domain (p.[Pro529Thrfs*17] and p.[Gln592*]) causing disease, even though the catalytic domains of GalNAc-T3 seem to be intact. Splice site mutations lead to exon skipping and cause LOF of the protein.

*Galnt3*-deficient or *N*-ethyl-*N*-nitrosourea (ENU)-induced GalNAc-T3^W589R^ (p.[Trp589Arg])-mutated mice show moderate phenotypes of hyperphosphatemic tumoral carcinosis with decreased alkaline phosphatase activity and intact FGF23 levels [77,78]. Interestingly, *Galnt3*-deficient mice do not develop ectopic calcification, which is a phenotype in human HFTC, but GalNAc-T3^W589R^ mice have this phenotype. Ichikawa et al. further showed that *Galnt3*-deficient male mice have growth retardation, infertility and increased bone mineral density, compared with wild-type mice [77]. These findings also occur in GalNAc-T3^W589R^ mice [78], indicating that GalNAc-T3 activity is important for male infertility, consistent with the high *GALNT3* expression in testis. 

Mandel et al. showed strong expression of GalNAc-T3, but not GalNAc-T1, in human spermatozoa [79]. The same group also showed GalNAc-T3 expression in spermatocytes and spermatids, where T antigen (Gal-GalNAc-Ser/Thr) is expressed. Importantly, GalNAc-T3 was the only GalNAc-T isoform expressed in ejaculated spermatozoa (although GalNAc-T17 may also be present; see below) and had ring-shaped equatorial expression, indicating that this isoform has a specific role in germ cells [80]. The fraction of spermatozoa with equatorial expression of GalNAc-T3 was significantly lower in men with oligoteratoasthenozoospermia. Therefore, GalNAc-T3 expression seems to be related to the quality of spermatozoa, and GalNAc-T3 deficiency may lead to impaired *O*-glycosylation of proteins and abnormal maturation and function of spermatozoa [81]. One candidate protein is zonadhesin, a cell membrane adhesion molecule that plays a role in sperm–egg binding and has a mucin-like tandem repeat region that is heavily *O*-glycosylated [82,83]. Miyazaki et al. [84] showed that spermatozoa in *Galnt3*-deficient mice are rare and immotile, and have deformed round heads. A glycoprotein localized at the equatorial segment of the acrosome, equatorin, was identified as another candidate target protein specifically glycosylated by GalNAc-T3, possibly at Thr138. This site of *O*-glycosylation on equatorin is important for sperm–egg interaction in vitro and in vivo [85,86]. *GALNT3* deficiency-related HFTC cases provided the first evidence for non-redundant functions of GalNAc-Ts in mammals. It is still unclear if HFTC males have reduced fertility, but *O*-glycosylation of sperm proteins by GalNAc-T3 seems to be important for sperm–egg interaction. 

### 2.4. GALNT4-Related Phenotypes

Murine *Galnt4* [16] is highly expressed in the sublingual gland, stomach and colon, and moderately expressed in the lung, small intestine, cervix and uterus. Human *GALNT4* [87] is highly expressed in the liver, small intestine, stomach, pancreas, thyroid, spleen, lymph node and bone marrow. GalNAc-T4^V506I^ (p.[Val506Ile]) carriers were shown to be associated with reduced risk of acute coronary syndrome (ACS), although the association was not significant in the haplotypic test [88].

### 2.5. GALNT5-Related Phenotypes

Rat *Galnt5* is highly expressed in the sublingual gland and colon, and moderately in the stomach and small intestine [89]. The partial human GalNAc-T5 amino acid sequence was published in 1999 [90]. *GALNT5*-related phenotypes have not been reported in mammals so far.

### 2.6. GALNT6-Related Phenotypes

Human *GALNT6* was identified as a close homolog of *GALNT3* [83]. Human *GALNT6* is expressed in the placenta and trachea, with weak signals in the brain and pancreas. Fibronectin is a good substrate for *O*-glycosylation by human GalNAc-T3 and GalNAc-T6, but not by GalNAc-T1 and GalNAc-T2, in vitro. However, GalNAc-T3 is not expressed in fibroblasts, in contrast to GalNAc-T6. These findings indicate that fibronectin is specifically glycosylated by GalNAc-T6 in vivo. Unlike GalNAc-T3, GalNAc-T6 is not expressed in human spermatozoa, whereas expression of *GALNT6*, but not *GALNT3*, occurs in the brain [83]. Related to this observation, Akasaka-Manya et al. showed that expression of several *GALNTs*, including *GALNT6*, is altered in sporadic Alzheimer’s disease progression in human brain. More prominent amyloid precursor protein (APP) *O*-glycosylation occurred through GalNAc-T6 compared to GalNAc-T1 or GalNAc-T4, and β-amyloid (Aβ1-40 and Aβ1-42) generation was reduced by overexpression of GalNAc-T6 without affecting the activities of secretases in vivo. This suggests that *O*-glycosylation of APP by GalNAc-T6 inhibits APP cleavage, and thereby decreases Aβ production [91]. Furthermore, downregulation of both GalNAc-T3 and GalNAc-T6 in the ectopic endometrium contributes to the development of endometriosis [92]. 

### 2.7. GALNT7-Related Phenotypes

Rat *Galnt7* (first reported as *Galnt6*) was identified by Ten Hagen et al. and human *GALNT7* was published by Bennet et al. in 1999 [90,93]. Rat and murine *Galnt7* are expressed in the sublingual gland, stomach, small intestine and colon, with trace amounts in the ovary, cervix and uterus. Human *GALNT7* is expressed ubiquitously, including in the stomach, thyroid, spinal cord, lymph node, trachea and adrenal gland, but not in bone marrow. This enzyme has a preference for GalNAc-glycosylated substrates and functions as a follow-up enzyme [90]. Genome-wide linkage analysis revealed that human *GALNT7* is positively associated with schizophrenia, although the association was not statistically significant in quantitative PCR analyses [94]. A GWAS showed that neuritic plaques, a core neuropathologic feature of Alzheimer’s disease formed by β-amyloid deposits, are positively associated with *GALNT7* [95]. Together with GalNAc-T6, GalNAc-T7 may control β-amyloid generation in Alzheimer’s disease. 

### 2.8. GALNT8-Related Phenotypes

Human *GALNT8* was cloned as an autosomal dominant hypophosphatemic rickets (ADHR) candidate gene from human fetal brain [96]. The tissue distribution of *GALNT8* is high in the heart, skeletal muscle, kidney and liver; moderate in the placenta, small intestine, leukocyte and lung; and weak in the brain, colon, thymus and spleen. *GALNT8* is highly polymorphic, but no substitutions are ADHR-related. The enzymatic activity of GalNAc-T8 was unclear until several studies showed that it might be a non-functional enzyme in vitro, despite weak expression of *GALNT8* in the fetal brain, testis, colon and small intestine found by quantitative real-time PCR, in contrast to a previous study [97,98].

### 2.9. GALNT9-Related Phenotypes

Human *GALNT9* shows brain-specific expression, especially in the cerebellum, frontal lobe, temporal lobe and putamen; weak expression in the cerebral cortex; and no expression in the medulla, occipital pole and spinal cord [99]. This distribution is consistent with quantitative real-time PCR results reported by Raman et al. [97]. Its enzymatic activity was shown by Zhang et al. [100] to be similar to GalNAc-T1 in terms of substrate specificities. GWAS in chickens showed that *GALNT9* in the liver differed significantly between fat and lean broilers, and correlated with abdominal fat traits, but it is unclear if this is also true in mammals [101].

### 2.10. GALNT10-Related Phenotypes

Rat *Galnt10* was identified (but referred to as *Galnt9* at the time) by Ten Hagen et al., together with *Galnt5* from rat sublingual gland [102]. The highest levels of rat *Galnt10* occur in the sublingual gland, testis, small intestine, colon and ovary; with lower levels in the heart, brain, spleen, lung, stomach, cervix and uterus. GalNAc-T10 resembles GalNAc-T7, in that it also serves as a follow-up enzyme and does not utilize non-glycosylated peptides as substrates. Murine *Galnt10* is predominantly expressed in several hypothalamic, thalamic and amygdalar nuclei in the brain. The broad expression of *Galnt10* in mouse brain might indicate that the enzyme has a role in the central nervous system (CNS) [103]. Human *GALNT10* was cloned as a homolog of human *GALNT7*. Quantitative real-time PCR revealed that human *GALNT10* is ubiquitously expressed, with the highest levels in the small intestine, and intermediate levels in the stomach, pancreas, ovary, thyroid gland and spleen. Human GalNAc-T10 displays strong activity for glycosylated peptides, but negligible catalytic activity toward non-glycosylated peptides [104,105]. Interestingly, a further study showed that the catalytic domain of GalNAc-T10 is involved in site selection on glycopeptides, but the lectin domain is not required for catalysis [106]. Expression of *Galnt10* in the brain in mice might be related to human schizophrenia, and a recent GWAS showed an unambiguous association between schizophrenia and *GALNT10* with other glycosyltransferases [107]. Together with GalNAc-T7, follow-up or glycopeptide-preferring GalNAc-Ts may be associated with serious mental diseases, but further work is needed to draw a clear conclusion.

### 2.11. GALNT11-Related Phenotypes

Human and murine *GALNT11* were identified in 2002 [5]. Human *GALNT11* is highly expressed in the kidney and moderately expressed in the brain, heart and skeletal muscle; in contrast, murine *Galnt11* is highly expressed only in the kidney. A recent GWAS identified an association between human *GALNT11* and deterioration of kidney function [108], and an assessment of GalNAc-T11 function in the kidney by Tian et al. [109] showed that *Galnt11*-deficient mice suffered from low-molecular-weight proteinuria. The same study also identified the endocytic receptor megalin/LRP2, a member of the low-density lipoprotein receptor (LDLR) family, as a specific substrate for GalNAc-T11 in the kidney. Megalin showed reduced binding to endogenous ligands in the absence of *Galnt11*. Another important aspect of *GALNT11* is that its expression is related to heterotaxy, a congenital heart disease resulting from abnormalities in left–right body patterning [110]. A case report of a 22-month-old male indicated that *GALNT11*, together with *GALNT20* (*GALNTL5*), which seems to be a non-functional protein (see below), was associated with developmental delay, distinctive facial features and multiple congenital anomalies [111]. These observations may be related to Notch signaling activation due to Notch1 *O*-glycosylation by GalNAc-T11 [112]. *Galnt11* and *NOTCH1* knockdown and gain-of-function produced similar left–right patterning in Xenopus, suggesting that *GALNT11* affects the Notch signaling pathway [112].

### 2.12. GALNT12-Related Phenotypes

Human *GALNT12*, which is most homologous to *GALNT4*, was cloned and shown to be expressed in digestive organs, including the stomach, small intestine, pancreas and colon. Moderate expression of *GALNT12* occurs in the testis, thyroid gland and spleen [113]. Recent GWASs have shown that *GALNT12* is negatively associated with serum galactose-deficient IgA1 levels resulting in IgA nephropathy [114], and a single nucleotide polymorphism (SNP)—rs2295926, belonging to *GALNT12*—is strongly associated with rapid radiographic joint destruction in patients with rheumatoid arthritis [115], but the underlying molecular mechanisms are unclear.

### 2.13. GALNT13-Related Phenotypes

Murine *Galnt13* was cloned and published as a homolog of *Galnt1* in 1995 [116]. T-cell specific depletion of the gene in mice resulted in no phenotype. Human *GALNT13* was cloned and shown to be specifically expressed in the brain [100]. In this study, *Galnt13* null mice were generated; they had a decrease in the Tn (GalNAc-Thr/Ser) antigen in Purkinje cell bodies and internal granular layer cells of the cerebellum cortex, compared with wild-type mice. However, the mice were fertile and developed normally. A case of a 13-year-old girl showed a relationship of *GALNT13* with minor facial and digital anomalies, mild developmental delay during infancy and behavioral disorders, which might reflect high expression of *GALNT13* in the brain [117]. 

### 2.14. GALNT14-Related Phenotypes

Human *GALNT14* was cloned based on *GALNT2* and is expressed specifically in the kidney [118]. Using autozygosity mapping and whole-exome sequencing, *GALNT14* with homozygous mutation in exon13 (c.[1273C>T]) resulting in truncation of >75% of the protein sequence (p.[Arg425*]) has been linked to embryonic lethality [119]. A heterozygous intragenic deletion of 123 kb in *GALNT14* was identified in an intestinal malrotation case by analysis of array comparative genomic hybridization (aCGH) data [120]. A whole-exome sequencing study revealed that a homozygous frameshift in *GALNT14* (c.[60del] or p.[Leu21Cysfs*6]) was identified as the genetic cause of Keratoconus in two cases [121]. However, the molecular mechanisms behind these observations are yet to be known.

### 2.15. GALNT15 (GALNTL2)-Related Phenotypes

Human *GALNT15* was cloned using cDNA of human cerebellum as a template. The transcripts were ubiquitously expressed in human tissues, with high expression in the placenta and small intestine, and moderate expression in the spleen, ovary and cerebral cortex [122]. Even though its enzymatic activity using peptide substrates was confirmed in vitro, *GALNT15*-related phenotypes in mammals have not been reported.

### 2.16. GALNT16 (GALNTL1)-Related Phenotypes

Human *GALNT16* was directly submitted to GenBank (AB078143.1), and the expression and catalytic activity of human GalNAc-T16 were studied by Raman et al. [97]. The enzyme transfers GalNAc to a wide range of peptide substrates. Human *GALNT16* has a broad distribution, with high expression in the heart and moderate expression in the brain and spinal cord. A recent GWAS reported an association with risk for cardiovascular disease [123]. In mice, *Galnt16* expression was very low in the tissues tested, including brain, colon, heart, kidney, liver, lung, skeletal muscle, ovary, prostate, spleen, sublingual gland, testis, thymus and thyroid [18 (as GalNAc-Ta)], but upregulation through an unknown mechanism was observed in diabetic mice [124].

### 2.17. GALNT17 (GALNTL6)-Related Phenotypes

Human *GALNT17* (described as *GALNT20* at the time) was cloned based on the *GALNT10* sequence, and showed high expression in the testis, brain and ovary [125]. In situ hybridization studies in mice indicated that *Galnt17* might have functions in spermatogenesis. GalNAc-T17 is also a follow-up GalNAc-T and prefers glycosylated peptides as substrates [125]. These results were reconfirmed by Raman et al. [97]. Recent GWASs have revealed that *GALNT17* (*GALNTL6*) polymorphism is associated with athletic performance through an unknown mechanism [126]. The data from the study suggested that *GALNT17* (*GALNTL6*) rs558129 T allele carriers had significantly higher power values in a Wingate anaerobic test than those with the CC genotype. Furthermore, the T allele was overrepresented in power athletes compared with both endurance athletes and controls. Therefore, the *GALNT17* (*GALNTL6*) rs558129 T allele could be favorable for anaerobic performance and strength of athletes [126]. A study in mice indicated that *Galnt17* dysregulation is related to a phenotype in human Autism Susceptibility Candidate 2 (AUTS2) syndrome, although the underlying mechanism is unclear [127].

### 2.18. GALNT18 (GALNTL4)-Related Phenotypes

Human GalNAc-T18 is an interesting enzyme that is primarily distributed in the endoplasmic reticulum (ER), rather than in the Golgi apparatus, where other GalNAc-Ts are located [98,128,129]. Human *GALNT18* is expressed ubiquitously, including in the lung, brain, uterus, placenta, testis and kidney [97,98], whereas mouse *Galnt18* is expressed only in the lung [18 (as GalNAc-T8)]. Human GalNAc-T18 has highly specific substrate specificity and only two peptides (GTTAKPTTLKPTE and GAGAEAPTPAPAGAGK) have been identified that can be glycosylated by the enzyme [97,98]. A recent longitudinal and transancestral study showed a positive association between demethylation of a CpG site located within *GALNT18* and development of active lupus nephritis [130]. In the study, a significant reduction in DNA methylation levels in a single CpG site (cg16204559) was observed during active nephritis in lupus patients. However, understanding the biological role of this demethylation in lupus nephritis will require further investigation [130].

### 2.19. GALNT19 (GALNTL3)-Related Phenotypes

Human and rat *GALNT19* were cloned and published by Nakamura et al. as homologs of human *GALNT9* [131]. As for *GALNT9*, human and rat *GALNT19* are highly expressed in the brain, and human *GALNT19* is also moderately expressed in the heart. Its enzymatic activities toward peptides are unclear [97, 98 (as GalNAc-T17), 131]. Interestingly, human *GALNT19* was found to be identical to the gene WBSCR17, located in the critical region of patients with Williams–Beuren Syndrome, a neurodevelopmental disorder. Therefore, GalNAc-T19 may glycosylate brain-specific substrates in vivo [131].

### 2.20. GALNT20 (GALNTL5)-Related Phenotypes

Human *GALNT20* is unique among *GALNTs* because it does not encode a lectin domain [97]. Highly specific expression of this gene in the testis has been detected in humans and mice [18,97]. Importantly, a heterozygous mutation of *Galnt20* affected male fertility with impairment of sperm motility; however, enzyme activity of GalNAc-T20 against peptides could not be detected in vitro [97,132]. This might indicate that the panel of peptide substrates used for detecting GalNAc-T activity is not sufficient, and that there is a need to find “true substrates” and include these substrates in future assays.

## 3. Conclusions

Almost 10 years have passed since the 20th human GalNAc-T was cloned and published. Some of these enzymes seem to have roles in vivo, but their enzymatic activities have yet to be shown in vitro. Recent GWASs have revealed human phenotype-related *GALNTs*, and these studies may also be able to identify true substrates for each GalNAc-T in vivo. GalNAc-Ts may also have functions other than as GalNAc-transferases, which would be particularly interesting. Recently, the function of *O*-glycosylation of the proteins initiated by GalNAc-Ts has been reviewed in a broader perspective [133]. Most *GALNTs* are associated with carcinogenesis, as widely shown, but we have not included these findings in this review because the studies were reviewed by Hussain et al. recently [134]. Recent studies targeting each *GALNT* by microRNA (miRNA) were also not included, because a single miRNA targets several messenger RNAs and the publications reporting the studies are still few, although this would be an appropriate subject for a further review.

## Figures and Tables

**Figure 1 molecules-26-05504-f001:**
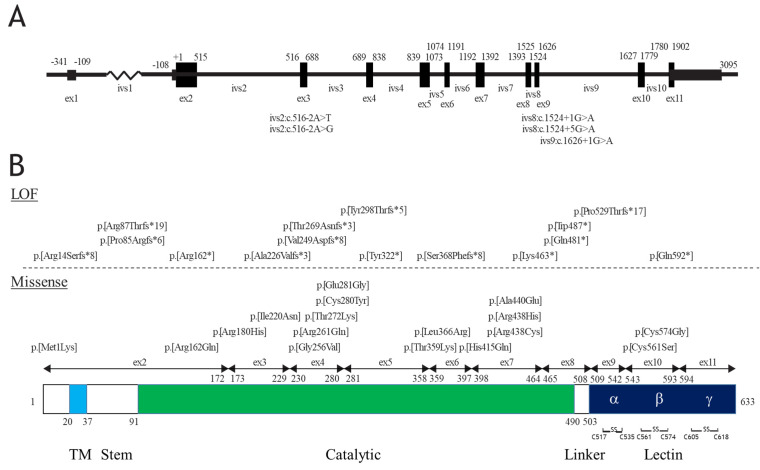
*GALNT3* and GalNAc-T3 mutations reported in HFTC patients. (**A**) Structure for the *GALNT3* transcript (NM_004482.4) with positions of exons in the transcript. Note that exon1 is non-coding. Splice site variants are shown with the position in the corresponding introns. (**B**) The protein domain structure of GalNAc-T3 with the exon positions and LOF and missense mutations is shown. ex; exon.

**Table 1 molecules-26-05504-t001:** Summary of published *GALNT2* mutations and associated phenotypes.

#	Nt Change ^(a)^	Exon	Consequence ^(a)^	Domain Affected	Genotype Status	Phenotype ^(1)^	Reference
1	c.[296dup]	3	p.[Tyr99*]	Stem	Homozygote	*GALNT2*-CDG	[43]
2	c.[311T>C]	3	p.[Phe104Ser]	Catalytic	Homozygote	*GALNT2*-CDG	[29]
3	c.[598C>T]	6	p.[Arg200*]	Catalytic	Homozygote	*GALNT2*-CDG	[43]
4	c.[629G>C]	7	p.[Arg210Pro]	Catalytic	Homozygote	*GALNT2*-CDG	[43]
5	c.[865C>T]	9	p.[Gln289*]	Catalytic	Homozygote	*GALNT2*-CDG	[29,43]
6	c.[914A>C]	10	p.[Asp314Ala]	Catalytic	Heterozygote/Homozygote	Affected HDL-C levels ^(2)^	[27,29]

(a) Reference sequences: NM_004481.5. and NP_004472.1. Nomenclature according to (https://varnomen.hgvs.org/ accessed on 1 May 2020); Mutation names according to Mutalyzer website (https://mutalyzer.nl/ accessed on 15 March 2021). (1) The phenotype for *GALNT2*-congenital disorder of glycosylation (CDGIIt) refers to OMIM #61885 and that for affecting HDL-C levels refers to sources [27,29]. (2) GalNAc-T2^D314A^ heterozygous mutation raised [27] and homozygous mutation lowered [29] plasma HDL-C levels in humans, for unknown reasons. #; number.

**Table 2 molecules-26-05504-t002:** Summary of published *GALNT3*-related HFTC cases.

Family	Alleles (NM_004482.4) (a)	AA Change (NP_004473.2) (a)	Ethnicity	dbSNP Id	# Family	# Cases	Genotype Status	Reference
1	c.[2T>A];[839G>A]	p.[Met1Lys];[Cys280Tyr]	Turkish	NA;NA	1	1	Compound heterozygote	[52]
2	c.[485G>A];[485G>A]	p.[Arg162Gln];[Arg162Gln]	Turkish	rs766717305	1	1	Homozygote	[49]
3	c.[539G>A;659T>A];[539G>A;659T>A]	p.[Arg180His;Ile220Asn];[Arg180His;Ile220Asn]	Chinese	rs772410242	1	2	Homozygote for 2 mutations	[47]
4	c.[767G>T];[767G>T]	p.[Gly256Val];[Gly256Val]	Caucasian, Norway (consang)	NA	1	2	Homozygote	[48]
5	c.[782G>A];[782G>A]	p.[Arg261Gln];[Arg261Gln]	Iranian	rs111875321	1	1	Homozygote	[53]
6	c.[815C>A];[1076C>A]	p.[Thr272Lys];[Thr359Lys]	Caucasian	rs137853090/rs137853091	1	1	Compound heterozygote	[54]
7	c.[842A>G];[1097T>G]	p.[Glu281Gly];[Leu366Arg]	Indian, Tamil	NA/rs780440401	1	2	Compound heterozygote	[55]
8	c.[1245T>A];[1245T>A]	p.[His415Gln];[His415Gln]	Turkish	NA	1	1	Homozygote	[56]
9	c.[1312C>T];[1774C>T]	p.[Arg438Cys];[Gln592*]	Caucasian (French/Canadian)	rs1159208891/rs137853087	1	1	Compound heterozygote	[57]
10	c.[1312C>T];[1774C>T]	p.[Arg438Cys];[Gln592*]	NA	rs1159208891/rs137853087	1	1	Compound heterozygote	[58]
11	c.[1312C>T];[1312C>T]	p.[Arg438Cys];[Arg438Cys]	Pakistani	rs764730691	1	1	Homozygote	[56]
12	c.[1313G>A];[1313G>A]	p.[Arg438His];[Arg438His]	Colombian	rs1159208891	1	1	Homozygote	[59]
13	c.[1319C>A];[484C>T]	p.[Ala440Glu];[Arg162*] (1)	Afro-American	rs199930810/rs137853086	1	1	Compound heterozygote	[60]
14	c.[1681T>A];[1681T>A]	p.[Cys561Ser];[Cys561Ser]	Indian	NA	1	1	Homozygote	[61]
15	c.[1720T>G];[1720T>G]	p.[Cys574Gly];[Cys574Gly]	Caucasian, Greek	rs267606841	1	1	Homozygote	[49]
16	c.[484C>T];[484C>T]	p.[Arg162*];[Arg162*]	African	rs137853086	1	1	Homozygote	[62]
17	c.[484C>T];[484C>T]	p.[Arg162*];[Arg162*]	African	rs137853086	1	1	Homozygote	[63]
18	c.[966T>G];[1441C>T]	p.[Tyr322*];[Gln481*]	Caucasian	rs137853088/rs137853089	1	2	Compound heterozygote	[64]
19	c.[1387A>T];[1387A>T]	p.[Lys463*];[Lys463*]	Caucasian, Italy	NA	1	2	Homozygote	[65]
20	c.[1460G>A];[1460G>A]	p.[Trp487*];[Trp487*]	Turkish	NA	1	1	Homozygote	[66]
21	c.[1774C>T];[1774C>T]	p.[Gln592*];[Gln592*]	Caucasian (Northern European)	rs137853087	1	1	Homozygote	[67]
22	c.[42_57del];[42_57del] (2)	p.[Arg14Serfs*8];[Arg14Serfs*8]	Caucasian	NA	1	1	Homozygote	[68]
23	c.[254_255del];[254_255del]	p.[Pro85Argfs*6];[Pro85Argfs*6]	Turkish	NA	1	2	Homozygote	[69]
24	c.[677del];[677del]	p.[Ala226Valfs*3];[Ala226Valfs*3]	Sri-Lanka	rs786205250	1	1	Homozygote	[49]
25	c.[746_749del];[892del]	p.[Val249Aspfs*8];[Tyr298Thrfs*5] (3)	NA	rs774681591;rs771510644	1	1	Compound heterozygote	[58]
26	c.[746_749del];[746_749del]	p.[Val249Aspfs*8];[Val249Aspfs*8]	Caucasian (Portuguese)	rs774681591	1	1	Homozygote	[70]
27	c.[1102dup];[1102dup]	p.[Ser368Phefs*8];[Ser368Phefs*8]	Iranian	NA	1	1	Homozygote	[66]
28	c.[1584dup];[1584dup]	p.[Pro529Thrfs*17];[Pro529Thrfs*17]	NA	NA	1	2	Homozygote	[58]
29	c.[516-2A>T];[484C>T]	p.[?];[Arg162*]	Afro-American	rs761396172/rs137853086	1	3	Compound heterozygote	[71]
30	c.[516-2A>T];[260_266del]	p.[?];[Arg87Thrfs*19]	NA	rs761396172/NA	1	2	Compound heterozygote	[58]
31	c.[516-2A>T];[1524+5G>A]	p.[?]:[?]	NA	rs761396172/rs375879489	1	1	Compound heterozygote	[58]
32	c.[516-2A>T];[516-2A>T]	p.[?]:[?] (5)	African	rs761396172	1	2	Homozygote	[72]
33	c.[516-2A>G];[516-2A>G]	p.[?]:[?]	Caucasian	rs761396172	1	1	Homozygote	[73]
34	c.[1524+1G>A];[1524+1G>A] (4)	p.[?]:[?]	Arabic-Lebanese	rs745655924	1	1	Homozygote	[49]
35	c.[1524+1G>A];[1524+1G>A]	p.[?]:[?] (6)	Arabic-Druze	rs760830864	1	6	Homozygote	[46]
36	c.[1524+1G>A];[1524+1G>A]	p.[?]:[?]	Arab-Israeli	rs760830864	1	2	Homozygote	[74]
37	c.[1524+1G>A];[1524+1G>A]	p.[?]:[?]	Jordan	rs760830864	1	1	Homozygote	[75]
38	c.[1524+5G>A];[484C>T]	p.[?];[Arg162*]	Afro-American	rs137853086/rs375879489	1	7	Compound heterozygote	[46]
39	c.[1626+1G>A];[803dup]	p.[?];[Thr269Asnfs*3] (7)	Caucasian (French)	rs766750282;rs766750282	1	1	Compound heterozygote	[76]
				Total	39 families	62 cases		

(a) Nomenclature according to <https://varnomen.hgvs.org/> accessed on 15 March 2021. Mutation names according to Mutalyzer website <https://mutalyzer.nl/> accessed on 1 May 2020; refseq: NM_004482.4. (1) Incorrectly named as A1319E and R484X [60]. (2) Named as nt position 41-58 [68]. (3) Incorrectly named as p.[Arg250Thrfs*2];[Tyr298Serfs*5] [47]. (4) Assigned as c.[1392+1G>A], but the chromatogram shows the sequence for ex8_ivs8 splice site region and the correct nomenclature for the mutation is c.[1524+1G>A]. (5) ex2 skipping was confirmed for c.[516-2A>T] allele. (6) ex8 skipping was confirmed for c.[1524+1G>A] allele. (7) ex9 skipping was confirmed for c.[1626+1G>A] allele.

**Table 3 molecules-26-05504-t003:** *GALNT3* mutations in HFTC and affected domains in GalNAc-T3.

#	Nt Change (NM_004482.4)	Exon	Consequence	Domain Affected	Alleles
1	c.[2T>A]	2	p.[Met1Lys]	Cytoplasmic	1
2	c.[485G>A]	2	p.[Arg162Gln]	Catalytic	2
3	c.[539G>A;659T>A]	3	p.[Arg180His];[Ile220Asn]	Catalytic	2
4	c.[767G>T]	4	p.[Gly256Val]	Catalytic	2
5	c.[782G>A]	4	p.[Arg261Gln]	Catalytic	2
6	c.[815C>A]	4	p.[Thr272Lys]	Catalytic	1
7	c.[839G>A]	5 splice site	p.[Cys280Tyr]	Catalytic	1
8	c.[842A>G]	5	p.[Glu281Gly]	Catalytic	1
9	c.[1076C>A]	6	p.[Thr359Lys]	Catalytic	1
10	c.[1097T>G]	6	p.[Leu366Arg]	Catalytic	1
11	c.[1245T>A]	7	p.[His415Gln]	Catalytic	2
12	c.[1312C>T]	7	p.[Arg438Cys]	Catalytic	4
13	c.[1313G>A]	7	p.[Arg438His]	Catalytic	2
14	c.[1319C>A]	7	p.[Ala440Glu]	Catalytic	1
15	c.[1681T>A]	10	p.[Cys561Ser]	Lectin	2
16	c.[1720T>G]	10	p.[Cys574Gly]	Lectin	2
17	c.[484C>T]	2	p.[Arg162*]	Catalytic	7
18	c.[966T>G]	5	p.[Tyr322*]	Catalytic	1
19	c.[1387A>T]	7	p.[Lys463*]	Catalytic	2
20	c.[1441C>T]	8	p.[Gln481*]	Catalytic	1
21	c.[1460G>A]	8	p.[Trp487*]	Catalytic	2
22	c.[1774C>T]	10	p.[Gln592*]	Lectin	4
23	c.[42_57del]	2	p.[Arg14Serfs*8]	Cytoplasmic	2
24	c.[254_255del]	2	p.[Pro85Argfs*6]	Stem	2
25	c.[260_266del]	2	p.[Arg87Thrfs*19]	Stem	1
26	c.[677del]	3	p.[Ala226Valfs*3]	Catalytic	2
27	c.[746_749del]	4	p.[Val249Aspfs*8]	Catalytic	3
28	c.[803dup]	4	p.[Thr269Asnfs*3]	Catalytic	1
29	c.[892del]	5	p.[Tyr298Thrfs*5]	Catalytic	1
30	c.[1102dup]	6	p.[Ser368Phefs*8]	Catalytic	2
31	c.[1584dup]	9	p.[Pro529Thrfs*17]	Lectin	2
32	c.[516-2A>T]	ivs2; ex2 skipping [72]	p.[?]	-	3
33	c.[516-2A>G]	ivs2	p.[?] or p.[Cys173Valfs*4]	-	4
34	c.[1524+1G>A]	ivs8; ex8 skipping [46]	p.[?]	-	8
35	c.[1524+5G>A]	ivs8	p.[?]	-	2
36	c.[1626+1G>A]	ivs9; ex9 skipping [76]	p.[?]	-	1
				Total number of alleles	78

## Data Availability

Not applicable.

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
