# Peer review of "Polypeptide N-acetylgalactosaminyltransferase-Associated Phenotypes in Mammals"

_molecules, 2021, doi:10.3390/molecules26185504_

Round 1

Reviewer 1 Report

The authors present an exceptional piece of science. They are key players in the field and have gathered a precise account of the current state of the art in the field along with the perspectives.

Author Response

Dear Reviewer,

Thank you very much for reviewing our manuscript.
Please see the attachment.

Best regards,
Kentaro Kato

Reviewer 2 Report

Dear editor,

This review manuscript by Kato et al summarizes the current literature on GALNT mutations and associated phenotypes in mammals. They described the history of GALNTs and detailed known phenotypes related to each GALNT. It was a good read for me to learn about the mutation’s effects of GALNT family. I recommend accepting this manuscript for publication. However, I would like to suggest to the authors if they could also add descriptions of mutations of GALNT that are involved in carcinogenesis. This would procure an additional value to the review.

Author Response

(The authors gave the same response as above.)

Reviewer 3 Report

The review article “GALNT mutation-associated phenotypes in mammals” provides an overview of the current literature on GALNT mutations and associated phenotypes in mammals. The major weak point of this manuscript is the lack of analysis and interpretation of the literature. The review looks like more a general description of evidence found in the literature rather than discussing the points themselves to provide a comprehensive picture of messages and/or flow of ideas to give to the reader. Before considering this manuscript for publication, major comments need to be revised as follows:

Detailed comments:

* The authors should consider changing the title and keywords and avoid the use of abbreviations.

* The abstract should be rewritten, it is poor and does not highlights the major points of the manuscript.

* Considering the focus of the manuscript, one scheme/table should be added indicating (for instance) the function/action/phenotype/genes involved on each GALNT mutation in mammals.

* Line 40: “GALNT1 and the protein are ubiquitously expressed in tissues and cell types at high levels [9, 10, 16-18]. This high and broad expression among tissues suggests that disruption of the gene would severely affect organisms. However, disruption of murine GALNT1 is not fatal and does not cause infertility, although GALNT1 deficiency does have moderate lethality and one-fourth of deficient mice die in utero beyond E12.5 or before the age of one month [19].” The authors should include an explanation why this happens.

* Define the abbreviations throughout the manuscript, e.g. ER stress, SMGs, …

* The topics 2.4 and 2.5 do not indicate the phenotypes associated with GALNT 4, GALNT5, GALNT14 and GALNT19 mutations, please provide more information or remove these parts.

* One figure should be added indicating the tissues where each GALNT mutation is highly expressed in mammals.

* Line 269: “This enzyme has a preference for GalNAc-glycosylated substrates and functions as a follow-up enzyme [89]. Genome-wide linkage analysis revealed that human GALNT7 is associated with schizophrenia [93] and a GWAS showed that neuritic plaques, a core neuropathologic feature of Alzheimer’s disease, are associated with GALNT7 [94].” The authors should provide additional explanations for this association. 

* Line 333: “These observations may be related to Notch signaling activation due to 333 Notch1 O-glycosylation by GalNAc-T11 [111].” The authors should explain this sentence. The same situation occurs in topics 2.12, 2.14, 2.17 and 2.18, it is necessary to add more discussion/information about these topics;

* Lane 422: “Recent studies of targeting each GALNT by micro RNA were also not included, but this would be an appropriate subject for a further review.” These studies should be provided considering their high relevance.

*The “micro RNA” word has an extra space.

Author Response

(The authors gave the same response as above.)

Reviewer 4 Report

Molecules 08/13/21

The review is a thorough assessment of the role of GALNTs in various biological processes across species. The information has been presented in a highly comprehensive manner and importantly, addresses the discrepancies in the field. The authors hint at other interesting functions of these enzymes which would be great topics for review as well.

Though much of this review is accurate and informative, there are a few scientific area(s) that need more attention:

Major:

  1. In page 2 line 60, the authors cite work mentioning that GALNT1 affects organogenesis in mice, yet the review mentions that GALNT1 knockout is not embryonic lethal. Is there a difference between homozygotes and heterozygotes of GALNT1 knockout or perhaps there is a compensating enzyme? Further explanation from literature would be beneficial.
  2. Page 3 from lines 100 to 128, the review discusses important contrasts in experiments. While this is very informative it leaves the reader confused. A sentence or two on what the authors think about these contradicting results would help guide the reader. Do the authors think that further clarification is needed, or do they believe the studies which state that GALNT2 increases HDL-C levels? In the same area, the overdose of AAV is concerning as the experiment should have a control to negate this issue. This statement needs further clarification.
  3. In lines 153 and 154, the authors mention that certain studies were omitted from the references used here. Was there a particular reason for the removal? Further information will be beneficial.
  4. Page 11, line 369 the authors mention the references which have studied the catalytic activity of GALNT6, but they do not mention the conclusion of this study. This needs to be incorporated in the text.
  5. In lines 393 and 394, it is mentioned that GALNT18 is highly specific and targets 2 peptides. Since it is only 2 peptides, more detail on the peptides would benefit the reader.
  6. In line 403, a reference for stating that GALNT19 might target brain-specific peptides needs to be included.
  7. The conclusion can be expanded to highlight interesting details of the statement regarding alternative functions of GALNTs. I found myself being very curious regarding them and this would further highlight the importance of GALNTs in cell signaling.
  8. One specific disease that is missing is cancer. GALNTs have been linked to cancer and it would help readers to have information regarding the role of each GALNT in cancer.

Minor:

  1. Overall, in the review, there appears to be a discrepancy in the nomenclature of the enzymes. It might be beneficial to stick to either GALNTs (italicized for gene) or GalNAc-Ts (italicized for gene).
  2. The sentence in lines 54 and 55 highlights the importance of GALNT1 in leucocyte rolling but would benefit from restructuring.
  3. The sentence from lines 348 to 351is informative but would benefit from restructuring. There are too many important points in the same sentence.
  4. In line 374 it is mentioned that “tissue” was tested. The nature of this tissue needs to be specified.
  5. Line 384 states that GALNT7 enhances athletic performance. This is very vague and needs clarification.
  6. Line 409 a very minor alteration, perhaps “affected” needs to be switched to “effected” or it can be switched to “affected male fertility”.

Overall, this review addresses an underrepresented knowledge base of GALNTs in biology, while emphasizing discoveries from various labs around the world in glycoscience. By highlighting the plethora of diseases that are dependent on GALNTs, the authors successfully argue the need to explore the biological function of GALNTs.

Author Response

(The authors gave the same response as above.)

Round 2

Reviewer 3 Report

I accept the manuscript in its present form.